# Structuring of Cold Pressed Oils: Evaluation of the Physicochemical Characteristics and Microstructure of White Beeswax Oleogels

**DOI:** 10.3390/gels9030216

**Published:** 2023-03-13

**Authors:** Sorina Ropciuc, Florina Dranca, Mircea Adrian Oroian, Ana Leahu, Georgiana Gabriela Codină, Ancuta Elena Prisacaru

**Affiliations:** Faculty of Food Engineering, Stefan cel Mare University of Suceava, 720229 Suceava, Romania

**Keywords:** oleogel, olive oil, grape oil, walnut oil, hemp seed oil, sunflower oil, physicochemical properties, microstructure

## Abstract

The aim of the study was to characterize the gelling effect of beeswax (BW) using different types of cold pressed oil. The organogels were produced by hot mixing sunflower oil, olive oil, walnut oil, grape seed oil and hemp seed oil with 3%, 7% and 11% beeswax. Characterization of the oleogels was done using Fourier transform infrared spectroscopy (FTIR), the chemical and physical properties of the oleogels were determined, the oil binding capacity was estimated and the SEM morphology was studied. The color differences were highlighted by the CIE Lab color scale for evaluating the psychometric index of brightness (L*), components a and b. Beeswax showed excellent gelling capacity at 3% (*w*/*w*) of 99.73% for grape seed oil and a minimum capacity of 64.34%for hemp seed oil. The value of the peroxide index is strongly correlated with the oleogelator concentration. Scanning electron microscopy described the morphology of the oleogels in the form of overlapping structures of platelets similar in structure, but dependent on the percentage of oleogelator added. The use in the food industry of oleogels from cold-pressed vegetable oils with white beeswax is conditioned by the ability to imitate the properties of conventional fats.

## 1. Introduction

Oleogels are useful alternatives for the structuring of vegetable oils without the use of trans or saturated fats. The structuring of oleogels in a three-dimensional network is achieved by supramolecular assemblies of gelling molecules and the retention of a large amount of oil in the structure [1]. Food technologies use solid fats in different recipes to obtain food products influencing the rheological, textural and sensory properties [2,3,4]. The use of oil in its initial form greatly limits the possibility of being used to obtain food products because liquid oil can negatively affect the texture of the products. In order to replace the solid fats used to obtain food products with oil structured in a viscoelastic network similar to a gel, the oleogelation technique was used [5]. Research has shown that the mechanical properties of oleogels beeswax (hardness or modulus of elasticity) can be affected by the characteristics of the vegetable oil used. It has been observed that the mechanical strength of the gels increases or decreases with the increase in the level of unsaturation of the oil. A canola oil gel rich in oleic acid (18:1) is much softer than a linseed oil gel rich in linolenic acid (18:3). Oil that is higher in terms of density will bind more tightly into the oleogel network, creating a stronger gel. Zetzl et al. (2012) demonstrated that EC (ethyl cellulose) gel hardness increases with increasing unsaturation for all oils tested [6].

During the process of formulating oleogels, the chemical composition of the oil and the nature of the gelling agent contribute significantly to obtaining oleogels with the desired structural, textural and rheological properties [7]. Edible vegetable oils contain dietary fatty acids, which are an indispensable nutritional resource for human health. Recently, with the increasing emphasis on healthy diets, the consumption of new vegetable oils with high nutritional value has been encouraged by the World Health Organization [7]. The purpose of using vegetable oils with unsaturated fatty acids instead of animal fat is to reduce cardiovascular, cerebrovascular diseases and mortality [8]. The oils extracted from woody plants, olive oil and walnut oil, contribute to numerous health benefits, such as supporting the elasticity of the arteries and implicitly helping to improve blood flow in the body, reducing both the chance of hardening of the arteries and the risk of cardiac diseases.

Cold pressed oils are accepted by consumers because they are organic, nutritious and safe food products. Due to the absence of heat treatment or any organic solvent in the extraction process, cold-pressed seed oils contain significant chemical qualities, high nutritional value, unique sensory characteristics and health-promoting elements. Cold pressed oils are rich in polyunsaturated fatty acids (PUFA), total monounsaturated fatty acids (palmitoleic acid, oleic acid, erucastic acid), total polyunsaturated fatty acids (linoleic acid, alpha-linoleic acid), chlorophylls, tocopherols, vitamin K and vitamin E, and by comparison with refined oils they present a state of autooxidation which that can be controlled through the oleogelation process [9,10]. The oxidation of cold-pressed oils is one of the major causes of the decrease in the nutritional value of food lipids, and it limits their preservation over time and leads to oxidative degradation products, volatile and non-volatile, which fundamentally change the organoleptic, nutritional qualities and innocuousness. It was shown that the type of oil has a significant influence on all characteristics of the oleogels. The use of different oils with the same technological treatment leads to the formation of crystals of diverse morphology [11].

Beeswax as a low molecular weight oleogelator presents a promising alternative for obtaining oleogels due to their ability to produce oleogelification at a minimum concentration of 3–5% [12]. This property of beeswax to form stable oleogels with attractive textural properties is due to its low polarity, longer chain length and high melting point [13,14,15,16]. Cold-pressed oils have different crystallization temperatures and melting points, which also influence the oleogelation property. Research on the morphology of beeswax gels describes the crystallization structure of oleogels in the form of platelets [17]. Many studies have described the possibility of obtaining oleogels by formulating with different gelling agents, but no study has described the formulation and analysis of the properties of gels obtained with walnut oil, hemp oil and grape seed oil using beeswax as a gelling agent [18]. Therefore, this study describes the characteristics of oleogels formulated with olive oil, grape seed oil, hemp seed oil, walnut oil and sunflower oil using white beeswax, added in a proportion of 3%, 7% and 11%.

## 2. Results and Discussion

### 2.1. Oleogel Characterization

The appearance of the oleogels is shown in Figure 1. All samples of oleogels formulated with beeswax (BW) and different vegetable oil assortments led to the formation of a stable gel. The oleogel samples showed good oil retention capacity in the structure, and no oil leakage was observed during the inversion of the containers at room temperature [19]. The oleogel tubes were removed from the refrigerator and kept at room temperature for 5 h, after which they were inverted to observe any oil leakage.

### 2.2. Oil Retention Test in the Oleogel Structure

The structuring of cold-pressed oils in a stable matrix indicates the effectiveness of the oleogelator used. The ability to retain oil in the structure is an important property of an oleogel, which represents the degree of capture of liquid oil in a stable network by the gelator [20,21,22]. Oleogels formulated with different vegetable oils can contain up to 99% (*w*/*w*) oil. The OBC values of the samples ranged from 64.34 to 99.73%. The lowest value was presented by the oleogel prepared with hemp seed oil and 3% BW. On the other hand, the highest value was observed at 11% BW using olive oils, grape seed oil, hemp seed oil and walnut oil. Similar results were obtained by Pandolsook and Kupongsak [23]. The retention of oil in the structure increased with the increase in the percentage of added wax. At the addition of 11%, the obtained oleogels presented a dense, saturated structure for all oil types used. After structuring the oleogel with wax percentages of 3% and 7%, the properties of the oleogels were improved by increasing the oil binding capacity [24,25]. In the present study, the oleogel formulated with 7% beeswax showed the best results for all types of oil because the obtained oleogels had a creamy, stable texture, without a rough, sandy structure. The results indicate that the specific gel structure was sufficiently homogeneous and stable to develop a network that holds the oil in a solid fat-like structure.

### 2.3. Peroxide Index Results

Peroxide index is the most common parameter used to characterize oils and fats in terms of oxidation and their validity period. The value of the peroxide index was determined 5 days after the formulation of the oleogels. They were kept in refrigerated conditions at a temperature of 5 °C. The peroxide index values are shown in Figure 2. The effect of the addition of beeswax as a structuring agent contributed to the stability of unsaturated fatty acid chains. [26,27]. All values were within the limit suggested by the Codex Alimentarius [28] for cold pressed and virgin oils (15 meqO_2_/kg). The oleogels formulated with olive oil (OL) showed the highest values, these values being correlated with the peroxide index of olive oil. Olive oil contains a high percentage of monounsaturated fatty acids (MUFA), which represents approximately 85% of its composition. The oleogels from walnut oil and hemp seed oil also had a high peroxide index [25,26]. The addition of beeswax significantly lowered the value of the oxidation index, which can be considered as a method of improving the oxidative stability for these categories of oils. With the increase in the beeswax dose, the index values decreased proportionally OL_11BW, WO_11BW and HO_11BW [29,30,31,32]. The fact that the wax changed the color of the oil, making it more opaque and less sensitive to light, can be used as a strategy to prevent oxidation. Another aspect is that the wax is lipophilic but also hydrophilic, thus generating a delayed oxidation reaction. Similar results were obtained in the formulation of oleogels intended for the production of spreads, margarines or animal fat vegetable substitutes [33,34,35].

### 2.4. Colorimetric Analysis

Color is an essential factor in determining the sensory qualities of food products that influence and support hypotheses about the taste and smell of products. Table 1 shows the average values of the color parameters for the analyzed oleogel samples. The CIE Lab color scale is considered a standard scale for evaluating the psychometric lightness index (L*) which ranges from black (0) to white (100). The a*component varies from green (negative) to red (positive), while the b*component varies from blue (negative) to yellow (positive) [30,31]. Consequently, the color scale is widely used in the food sector. The obtained results demonstrate the fact that the values regarding the yellow-red or green spectrum of the oleogel samples show close values. The brightness decreases due to the formation of fat nanoparticles that give the products a matte appearance. Hemp seed, olive and sunflower oils have high values of the negative component (a), this aspect is explained by the specific color of these oils that tends towards green. By comparison, grape seed oil has the lowest values of component a*Oleogels from walnut oil (WO) and sunflower oil (SO) tend towards colors in the yellow-red spectrum. The color difference quantified by the ΔE* value demonstrates that the samples have very small color variations between the samples obtained with the same type of oil. Even if the brightness of the oleogels decreases, their color with the addition of beeswax as an oleogelator is preserved depending on the color of the oil.

### 2.5. Scanning Electron Microscopy (SEM)

The micrographs are shown in Figure 3 and illustrate that the very thin, plate-like crystals formed in oil are platelets. The microstructure of the oleogels was examined by SEM microscopy. Figure 3 shows the images obtained by electron scanning microscopy, which allow a better view of the crystal network, assuming that the surface of the grid and its roughness are the main factors in determining the structural capacity of the oil [23]. The SEM microscopy allowed the visualization of platelet formations, the way the oil binds depending on the type of oil and the percentage of added wax [36]. The oily phase can be distinguished by dark areas, while the components that have formed gel formations, oil bound with beeswax, can be seen in a shiny form [37]. The appearance and structure of oleogels are given by the formation of bonds between wax and oleogel. Olive oil forms a fine network with a uniformly linked structure, without large pores. Grape seed oil (GO) produces different oleogel crystal morphologies than olive oil. Rough, granular formations can be observed in the microstructure, and the way the wax is bound is different. Walnut oils (WO) at the addition of 3% BW form large granular structures with oil expulsion. The same trend of agglomeration in large, overlapping structures is also noticeable in hemp seed oil (HO) [29,30,38]. The increase in the percentage of wax also determines the growth of crystal agglomerations in the network. In general, lower concentrations of BW generate larger crystal aggregates and changes are evident in the microstructure of the oleogels, especially for walnut oil. 

### 2.6. FTIR Spectra of Oleogels

The FTIR spectra of the oleogels obtained with white beeswax and different types of oils (olive oil, grape oil, walnut oil, hemp seed oil and sunflower oil) are shown in Figure 3. The region covered by the FTIR analysis was 4000 cm^−1^ and 650 cm^−1^ and contains information from molecular vibrations specific to the chemical composition of the samples, which is useful to gain insight into how the beeswax and oil combine to form the oleogel matrix and also to evaluate the effect of different edible oils on the crystallization of beeswax in oleogels.

As the spectra in Figure 4 shows, all samples presented similar absorption bands. For each oil type used to obtain the oleogels, the stacked spectra are shown in Figure 4a–e (the overlapped spectra are presented as Appendix A). According to previous studies, the oleogelator (beeswax) has the propensity to structure the solvent (vegetable oil) through a hierarchical molecular self-assembly via non-covalent interactions, resulting in the formation of a three-dimensional network [39,40]. As the oleogel forms, three main different spectral bands are observed in the FTIR spectra, which are assigned to O–H stretching vibrations, vibrations of the carboxyl group and asymmetric stretching vibrations of CH_3_ and CH_2_ groups. The band corresponding to O–H stretching modes is usually found around 3470 cm^−1^; as this spectral band was not observed for our samples, it can be concluded that no hydrogen bonds were formed in the oleogels with beeswax, which is in agreement with [41,42]. The small peak around 3009 cm^−1^ was assigned to the C–H stretching vibration of the *cis*-double bonds (=CH) of edible oils [43], while the pronounced peaks at 2921 cm^−1^ and 2852 cm^−1^ were attributed to symmetric and asymmetric stretching vibration of the aliphatic CH_2_ group. These may be characteristic to waxes, and were reported for various types of oleogels such as soybean oil-carnauba wax oleogel [44], shellac wax emulsion oleogels [45] and oleogels containing sunflower wax and sunflower oil [46].

Another prominent peak was identified in the spectra of the oleogel samples at 1743 cm^−1^ and was attributed to C=O stretching vibrations of esters and free fatty acids which result from the overlapping combination between oil and beeswax. Of interest in the case of this peak was the decrease in its intensity, as determined by the increase in the proportion of oil in the oleogel; the decrease was particularly noticeable for oleogels containing olive oil and hemp seed oil. The reduction of the peak at 1743 cm^−1^ may be due to interesterification, as a previous study also observed for walnut oil oleogels [47]. Furthermore, for oleogels containing hemp seed oil (Figure 4b), the increase in oil proportion led to an increase in the absorbance at 1726 cm^−1^; this peak was attributed to C=O stretching vibrations in COOH [48]. Previous studies reported that the peak at 1743 cm^−1^ reflects the composition in high molecular weight esters [49], and considering that beeswax has a high content in these esters, the reduction in the peak at 1743 cm^−1^ can be considered an indicator of the increase in oil together with the decrease in beeswax proportion. 

In the region between 1600 and 1000 cm^−1^, many differences were observed in the spectra and were determined by the increase in oil proportion in oleogels. All samples had peaks around 1457 cm^−1^, 1376 cm^−1^, 1160 cm^−1^ and 1098 cm^−1^, which corresponded to CH_2_ and CH_3_ bending, C–H symmetric bending vibrations of CH_2_, C–O stretching of esters, and –CH deformation vibrations of fatty acids [44]. The small peak at 720 cm^−1^ was due to CH_2_ rocking vibration, and can highlight the role of the wax in oleogels formulation. High absorbances at 720 cm^−1^ were recorded for the walnut oil oleogels in correlation with the microstructure analysis results.

### 2.7. Principal Component Analysis (PCA)

PCA (Figure 5) was performed to better characterize the relationships between the analyzed variables: oil binding capacity (OBC), lipid oxidation (PV) and color. The oleogel samples were divided into different quadrants depending on the correlation coefficient that indicates the strength of the bonds between them. Two main components (PC2/PC1) were extracted from the statistical analysis, which explained the total variance of the data set in a percentage of 75.21% respectively 29.59%. Oleogels with 3% BW correlate with the oxidation index and oil retention. The oleogels with 11% BW correlate with the brightness, and the oil retention capacity in the structure (OBC) correlates with oleogels that have the highest percentage of wax in the structure, respectively 11%. The oleogels were grouped according to the type of oil and the percentage of wax. The oleogels with grape seed oil and hemp seed oil are strongly correlated with the addition of 11% oleogelator. The same trend can be observed in the oleogels obtained with walnut oil and wax percentages of 3% and 7%. The olive oil oleogel with 3% wax (OL_3BW) correlates weakly with the other oleogels. This distribution is explained by the oil retention power in the weak structure, reduced stability and high value peroxide index.

## 3. Conclusions

In this study, the potential of using different types of vegetable oils for the formulation of oleogels with beeswax in animal fat replacement was explored. An important criterion in the formulation of oleogels is the oil binding capacity in the structure. The wax was dissolved at a temperature of 80 °C, the slowly cooled at a temperature of 40 °C and stored under refrigeration conditions. Oleogels with well-bound oil in the structure were obtained for most of the oils used. The analysis of the microstructure highlighted the gelation quality and the appearance of the oleogel at the level of microcrystals. FTIR spectra described the same binding behavior in a stable oil network with different proportions of wax. The results achieved can be considered directions for new research, suggesting possibilities for using oleogels from oil of: walnut, hemp seed, olive and sunflower in various food products. The research aimed to evaluate the physicochemical, structural and microstructure characteristics of oleogels obtained from different cold-pressed oils and white beeswax. The results regarding the ability to form a stable gel with a specific consistency, and with a peroxide index inside the limits provided by the legislation in force, show that the percentage of 7% wax added as an oleogelator for all types of oil is considered to be ideal.

## 4. Materials and Methods

### 4.1. Materials

#### Preparation of Oleogels

The experimental material consisted of oleogels that were produced using commercial vegetable oils: olive oil (OL), grape oil (GO), walnut oil (WO), hemp seed oil (HO), sunflower oil (SO) and the following structuring agent. All oils were obtained by cold pressing the oil seeds and were purchased from companies specializing in natural products [50]. The gelling agent used was white beeswax (BW) (Sigma-Aldrich, Hamburg, Germany). The characteristics of beeswax, according to the technical sheet: white, pure, natural beeswax, solid physical condition, waxy form, odorless, melting point 61–65 °C, density 0.96–0.98 g/cm³ at 20 °C, flammability—this material is combustible, but will not light up easily. The concentration of structuring substances in the oily solvent, determined by the scientific literature, was as follows: 3%, 7% and 11% *w*/*w*. The oil and wax were heated on an electric hob equipped with a magnetic stirrer until the wax completely dissolved. The heating temperature was a maximum of 80 °C while stirring at the speed of 500 rpm with the help of a multipoint magnetic stirrer (TA Instruments, New Castle, DE, USA), Figure 6. After the complete dissolution of the wax, the mixture of wax and oil was left at room temperature for tempering, then they were poured into glass tubes and kept for 5 days at a temperature of 5 °C for stabilization. The total number of samples formulated for the beeswax and vegetable oils was fifteen.

### 4.2. Methods

#### 4.2.1. Oil Binding Capacity (OBC)

To determine the oil binding capacity in the structure (OBC), 1.0 g of oleogel was placed into a 5 mL Eppendorf tube. The tubes were labeled and refrigerated at 4 °C for one hour. The samples were then centrifuged for 15 min at 4000 rpm and a temperature of 24 °C. For this purpose, a microcentrifuge (Hermle Z206A) with thermostat was used. After 15 min, the tubes were inverted for 5 min on a paper towel to remove the liquid fraction [22,33].

The following calculation formula was used:% OBC= (A − B)/ (C − B) × 100 (1)
in which:

A—tube and sample mass after centrifugation and removal of the liquid fraction [g]; B—tube weight [g]; C—tube and sample mass after centrifugation [g].

#### 4.2.2. Determination of the Peroxide Index (PV)

The determination of the oxidative stability of the oleogels was carried out through a chemical analysis test, by the standard methodology ISO 3960:2007. The test was carried out on the oleogel samples five days after obtaining them. The evaluation of the oxidation process was carried out according to the ISO 3960:2007 standard methodology [33,35,51]. The peroxide index value was calculated using the following formula:PV (meqO_2_/kg) = ((V1 − V0) × M × 1000 × T)/m (2)
where: V1—milliliters of Na_2_S_2_O_3_ solution required for the titration of the samples, [mL]; V0—milliliters of Na_2_S_2_O_3_ solution required for the titration of the blank sample [mL]; M—molarity of the Na_2_S_2_O_3_; T—0.0246 solution titer of Na_2_S_2_O_3_; m-sample table taken into work [g].

#### 4.2.3. Color Measurement of Oleogel

The color determination was carried out using the CIELab system (Konica-Minolta 200, Tokyo, Japan). The method is based on the representation of color, using three points L*, a*, b* [23,52]. The oleogels were placed in transparent Petri plaques which were then moved on a white surface. Numerical values of the L* brightness, as well as parameters a* and b*, were obtained indicating the color of oleogels. Color measurement takes place based on the X, Y, Z color tristimulus values that can be represented in Cartesian or cylindrical coordinates. The data were also used to determine the absolute color difference ∆E.

#### 4.2.4. Scanning Electron Microscopy (SEM)

The oleogels were subjected to morphological analysis with a scanning electron microscope. The oleogel samples were previously subjected to lyophilization, obtaining xerogels. This treatment allowed the removal of the aqueous phase from the structure of the oleogels. The xerogel was placed on a graphite plate. For imaging, an electron microscope was used with scanning TESCAN Vega II LMU Czech Republic, whose source of electrons—the cathode tube, generates the electrons by directly heating the tungsten cathode filament (working distance = 5 mm; acceleration voltage = 10 kV). This technique allows the characterization of fats and lipid crystals, their distribution, size and shape.

#### 4.2.5. FTIR Spectroscopy Analysis

The spectra of oleogel samples were recorded in the mid-infrared region of 4000–650 cm^−1,^ with a resolution of 4 cm^−1^, using a Nicolet iS-20 spectrometer (Thermo Scientific, Karlsruhe, Dieselstraße, Germany). Each sample was placed on the ATR surface and the spectra collection was made at 25 °C in duplicate using OMNIC software (version 32, Thermo Scientific). OMNIC Specta software was used to further process and display the collected spectra.

#### 4.2.6. Statistical Analysis

The obtained results were statistically analyzed with the Software XLSTAT, version 2021 (Addinsoft, New York, NY, USA), using one-way analysis of variance followed by the Tukey test for the determination of the significant differences between means at *p* < 0.05. Principal component analysis (PCA) was applied to observe the distribution of data sets obtained in the determination of physicochemical parameters.

## Figures and Tables

**Figure 1 gels-09-00216-f001:**
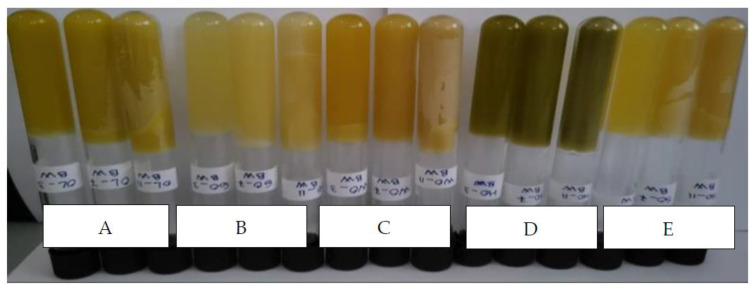
Visual appearance of oleogel with beeswax and vegetable oil. **Group A**—oleogels obtained from olive oil and 3%, 7% and 11% beeswax (OL_3BW; OL_7BW; OL_11BW); **Group B**—oleogels obtained from grape oil and 3%, 7% and 11% beeswax (GO_3BW; GO_7BW; GO_11BW); **Group C**—oleogels obtained from walnut oil and 3%, 7% and 11% beeswax (WO_3BW; WO_7BW; WO_11BW); **Group D**—oleogels obtained from hemp seed oil and 3%, 7% and 11% beeswax (HO_3BW; HO_7BW; HO_11BW); **Group E**—oleogels obtained from sunflower oil and 3%, 7% and 11% beeswax (SO_3BW; SO_7BW; SO_11BW).

**Figure 2 gels-09-00216-f002:**
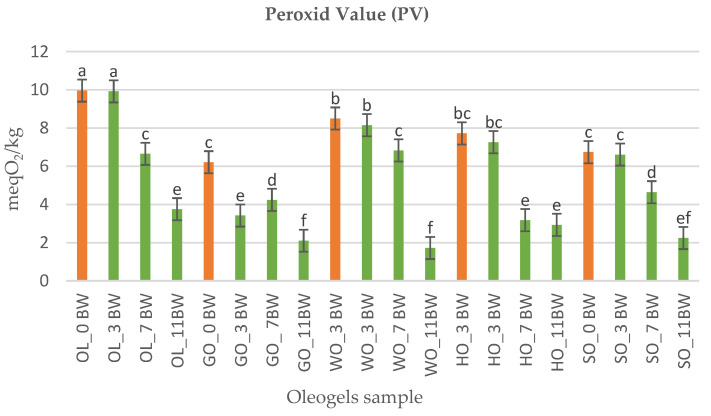
The graphic representation of peroxide index value. Different superscript letters in the same column indicate significant difference between values at *p* < 0.05 level.

**Figure 3 gels-09-00216-f003:**
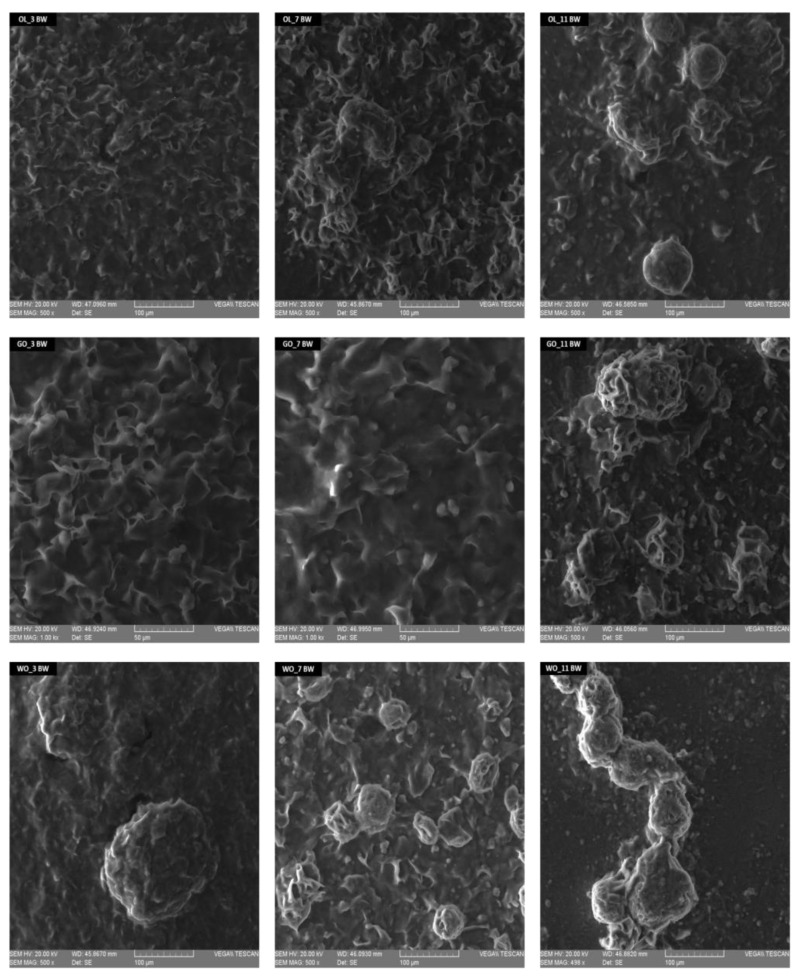
Scanning electronic microscopy (SEM) images of oleogels. Magnification of 500×. Scale equivalent to 50–100 μm.

**Figure 4 gels-09-00216-f004:**
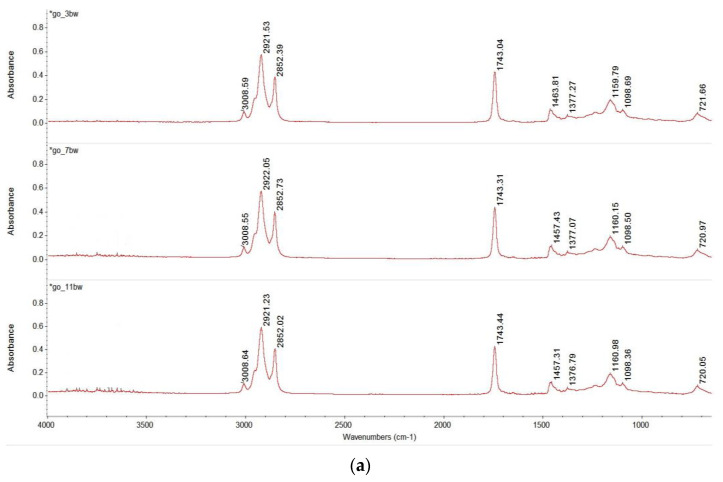
FTIR spectra of oleogels: stacked spectra of oleogels with (**a**) grape oil, (**b**) hemp seed oil, (**c**) olive oil, (**d**) sunflower oil and (**e**) walnut oil.

**Figure 5 gels-09-00216-f005:**
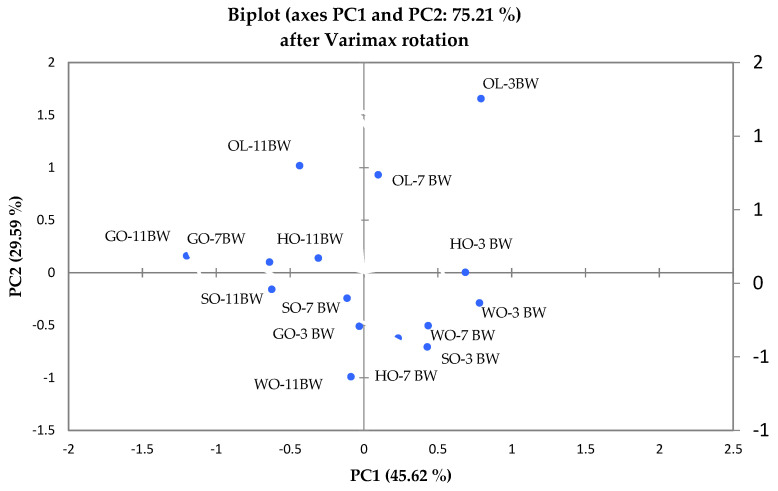
Graphical representation of principal component analysis between the oleogel samples.

**Figure 6 gels-09-00216-f006:**
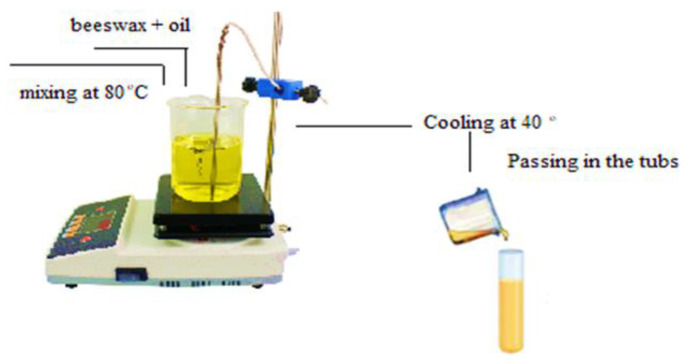
The method of obtaining the oleogel.

**Table 1 gels-09-00216-t001:** The results of oil binding capacity (OBC), stability and color analyses.

SAMPLE	OBC, %	STABILITY	L*	a*	b*	ΔE*
OL_3 BW	68.85 ± 0.11 ^d^	-	41.88 ± 0.04 ^b^	−5.19 ± 0.02 ^d^	32.60 ± 0.04 ^a^	59.95 ± 0.01 ^a^
OL_7 BW	76.23 ± 0.22 ^c^	+	45.67 ± 0.03 ^b^	−5.64 ± 0.05 ^d^	29.82 ± 0.02 ^b^	54.64 ± 0.04 ^b^
OL_11 BW	96.56 ± 0.16 ^a^	+	49.54 ± 0.01 ^ab^	−5.91 ± 0.03 ^d^	36.19 ± 0.04 ^a^	55.27 ± 0.02 ^b^
GO_3 BW	65.12 ± 0.12 ^d^	-	44.28 ± 0.02 ^b^	−3.17 ± 0.04 ^b^	18.14 ± 0.03 ^d^	51.84 ± 0.02 ^c^
GO_ 7 BW	82.54 ± 0.27 ^b^	+	51.89 ± 0.04 ^a^	−3.12 ± 0.04 ^b^	23.48 ± 0.02 ^c^	48.80 ± 0.04 ^d^
GO_11 BW	99.73 ± 0.16 ^a^	+	55.46 ± 0.03 ^a^	−3.38 ± 0.03 ^b^	27.08 ± 0.03 ^b^	46.28 ± 0.03 ^d^
WO_3 BW	64.41 ± 0.21 ^d^	-	42.88 ± 0.02 ^b^	−2.37 ± 0.04 ^a^	27.49 ± 0.05 ^b^	57.07 ± 0.03 ^a^
WO_7 BW	82.46 ± 0.15 ^b^	+	41.03 ± 0.02 ^b^	−2.82 ± 0.05 ^a^	25.67 ± 0.04 ^b^	54.61 ± 0.02 ^b^
WO_11 BW	98.34 ± 0.17 ^a^	+	41.34 ± 0.04 ^b^	−3.63 ± 0.01 ^b^	21.03 ± 0.03 ^c^	55.43 ± 0.01 ^b^
HO_3 BW	64.34 ± 0.11 ^d^	-	39.78 ± 0.02 ^c^	−4.55 ± 0.03 ^c^	26.92 ± 0.02 ^b^	59.24 ± 0.04 ^a^
HO_7 BW	83.98 ± 0.19 ^b^	+	39.99 ± 0.03 ^c^	−4.03 ± 0.02 ^c^	22.08 ± 0.03 ^c^	57.17 ± 0.02 ^a^
HO_11 BW	96.27 ± 0.18 ^a^	+	44.04 ± 0.02 ^b^	−5.83 ± 0.05 ^d^	25.54 ± 0.04 ^b^	54.75 ± 0.02 ^b^
SO_3 BW	68.45 ± 0.23 ^d^	-	41.47 ± 0.01 ^b^	−4.32 ± 0.04 ^c^	18.76 ± 0.02 ^d^	54.33 ± 0.04 ^b^
SO_7 BW	72.65 ± 0.27 ^c^	+	46.47 ± 0.02 ^b^	−5.12 ± 0.02 ^d^	25.53 ± 0.03 ^b^	52.23 ± 0.03 ^c^
SO_11BW	92.87 ± 0.20 ^a^	+	49.91 ± 0.04 ^ab^	−4.93 ± 0.01 ^cd^	28.77 ± 0.02 ^b^	50.63 ± 0.02 ^c^

Note: (−) unstable; (+) stable. Different superscript letters in the same column indicate significant difference between values at *p* < 0.05 level.

## Data Availability

Not applicable.

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
