# Peer review of "Structuring of Cold Pressed Oils: Evaluation of the Physicochemical Characteristics and Microstructure of White Beeswax Oleogels"

_gels, 2023, doi:10.3390/gels9030216_

Round 1

Reviewer 1 Report

This paper characterize the gelling effect of beeswax (BW) using different types of cold pressed oil, including oil absorption, POV, the color differences as well as structure and morphology by FTIR and SEM. It is meaningful and suitable for publication in Gels. However before accept would be suggest, ift would be better if following concerns could be addressed.

1. pressed oil composition is complex its lipid composition,fatty acid profile,unsaponifiable matter (VE, sterols etc.),initial moisture and free acid content all have impact on gellation and oxygenation. Hence, discussion should performed with respect to those factors.

2. As oil retention test experimental results, under high BW adding levels, gellation was predominate by structuring agent, while low BW cases, lipid composition effects could not be neglected, i.g sitosterol in rude oil, saturation degree, they all account for the differences for oil absorption among various oil type.

3. For POV, except oleogel, fatty profile, VE and squalene level may be are more important in term of oxygenation for oleogel prepared with different oil.

4. L229-230, oil level increased, absorption at 1740cm-2 decreased,why?

5. L234,  what does high molecular weight esters refer to  ? was not the absorption at 1740cm-2  assigned to the C=O in triglyceride ?

6. Figure 4, the absorption at 1720 band was only found in Fig 4b (Hemp oil gel), could you give a related discussion.

7. FTIR Spectroscopy analysis

How did you prepare samples for ATR ?  Oleagel is viscous and semisolid, was oleagel directly posed an surface of ATR, or melted or solved in solvent? Or how did you control the thickness of the oleagel samples , which is related with the FTIR absorption intensity.

8. At last, What characteristics of oil are suitable for oleagel preparation?

Author Response

Dear reviewer
I am attaching the revised article and cover letter
With many thanks,

Reviewer 2 Report

1. In the Abstract section, L17, the value 64.35% is inconsistent with the data in the Table 1. Please check.

2. In the Introduction section, L44, the abbreviation “EC gel” should be defined in the text.

3. In the section of 2.2. Oil retention test in the oleogel structure, L104, the sentence “The lowest value was presented by the oleogel prepared with walnut oil and 3% BW.” is not very precise, as the OBC value (64.34%) of hemp seed oil gel with 3% BW is a little bit lower than that of walnut oil gel.

Similarly, L105-1-6, the same description in the sentence “On the other hand, the highest value was observed at 11% BW using olive oils, grape seed oil and hemp seed oil”. How about the walnut oil with 11% BW?

Moreover, L111-112, “In the present study, the oleogel formulated with 7% beeswax showed the best results for 111 all types of oil”. What does this sentence mean?

4. In the section of 2.3. Peroxide index results, L126-127, “All values were within the limit suggested by the Codex Alimentarius [25] for cold pressed and virgin oils (15 meqO2/kg).” However, the order of magnitude of Y-axis seems incorrect. Please check.

Author Response

(The authors gave the same response as above.)

Reviewer 3 Report

Line 25: Please rewrite this section to be more clear and precise manner.

Line 81: Please add one paragraph at the end of this section to discuss the study's inadequacies and possible future work that would be warranted.

Line 117: Please include additional related work to strengthen the current study, such as; doi.org/10.1016/B978-1-893997-73-8.50008-6, doi.org/10.1016/j.bcab.2021.102122

Line 319: This section should be reduced and the related work should be expanded; for example, doi.org/10.1007/s11694-017-9514-5, doi.org/10.1007/s13197-018-3336-2, doi.org/10.1080/14786419.2014.901319

Line 343: Please elaborate on the most recent work in this segment; for instance, doi.org/10.1016/j.jfoodeng.2012.08.039, 10.1016/j.ijbiomac.2016.01.011, doi.org/10.1016/j.mimet.2006.11.008

Author Response

(The authors gave the same response as above.)
